# Donor Heart Preservation: Current Knowledge and the New Era of Machine Perfusion

**DOI:** 10.3390/ijms242316693

**Published:** 2023-11-24

**Authors:** Dimitris Kounatidis, Vassiliki Brozou, Dimitris Anagnostopoulos, Constantinos Pantos, Athanasios Lourbopoulos, Iordanis Mourouzis

**Affiliations:** Department of Pharmacology, National and Kapodistrian University of Athens, 11527 Athens, Greece; dimitriskounatidis82@outlook.com (D.K.); vasiliki.brozou@gmail.com (V.B.); dimitsius@gmail.com (D.A.); cpantos@med.uoa.gr (C.P.); alourbop@gmail.com (A.L.)

**Keywords:** cardioplegia, static cold storage, heart transplantation, hypothermic machine perfusion, kinases, normothermic machine perfusion, Organ Care System, thyroid hormone

## Abstract

Heart transplantation remains the conventional treatment in end-stage heart failure, with static cold storage (SCS) being the standard technique used for donor preservation. Nevertheless, prolonged cold ischemic storage is associated with the increased risk of early graft dysfunction attributed to residual ischemia, reperfusion, and rewarming damage. In addition, the demand for the use of marginal grafts requires the development of new methods for organ preservation and repair. In this review, we focus on current knowledge and novel methods of donor preservation in heart transplantation. Hypothermic or normothermic machine perfusion may be a promising novel method of donor preservation based on the administration of cardioprotective agents. Machine perfusion seems to be comparable to cold cardioplegia regarding donor preservation and allows potential repair treatments to be employed and the assessment of graft function before implantation. It is also a promising platform for using marginal organs and increasing donor pool. New pharmacological cardiac repair treatments, as well as cardioprotective interventions have emerged and could allow for the optimization of this modality, making it more practical and cost-effective for the real world of transplantation. Recently, the use of triiodothyronine during normothermic perfusion has shown a favorable profile on cardiac function and microvascular dysfunction, likely by suppressing pro-apoptotic signaling and increasing the expression of cardioprotective molecules.

## 1. Introduction

In recent years, an increase in the incidence of heart failure has been observed, a phenomenon attributed to the population aging. Despite progress in pharmaceutical and interventional treatment, heart transplantation remains the gold standard therapy for the treatment of end-stage heart failure. Heart transplantation is a life-saving strategy with favorable outcomes for most recipients despite its technical challenges. By the application of this method, the annual graft survival is 85–93%, while long-term survival is up to almost 70% in a ten-year period [1]. Heterogeneous factors have augmented the difficulties in its utilization in its half-century history, while the number of the available grafts cannot meet current needs.

Brain-dead donors (DBD) are the main source of hearts that are currently used for transplantation, affecting, in a negative way, the graft response. Organ dysfunction, structural alterations, inflammation, and infections are the main complications that can be seen [2]. On the other hand, the possibility of using hearts from circulatory determined death (DCD) donors has been relatively recently re-evaluated by the International Congress on Non-Heart Beating Donation, especially in pediatric patients. The main disadvantages of this strategy are the inability to assess the functionality of the asystolic heart and the unquantifiable warm ischemic injury of the myocardium.

Nevertheless, technological advances have led to the establishment of DCD heart transplant programs with encouraging outcomes. The beneficial results of using these grafts in kidney transplantation is the bright example for their utilization in heart transplantation as well. In recent years, an attempt has been made to overcome the problems that arise in the post-transplantation period by using post-conditioning pharmacological agents on an ex situ perfusion platform, allowing, at the same time, for the ability to determine potential biomarkers that may be used to predict primary graft dysfunction (PGD). Thus, preliminary evidence support that the use of DCD grafts may benefit heart failure patients, with at least equivalent results to DBD heart transplantation [3].

The limited number of available grafts and the need to extend donor selection forces the acceptance of a growing number of marginal grafts. On the other hand, discordance between the increasing need for transplants and organ availability, as well as population aging, extensive resuscitation, and shift in the causes of death, which are mirrored in the organ pool, elucidated the need to focus efforts on extending the latter as well as better preserving grafts [4]. As a result of that effort, current practice optimizes the support of the donor heart through donor management goals. Specific characteristics regarding evaluation of marginal donors’ acceptance can be seen in Table 1. Furthermore, current strategies of preserving the cardiac graft (static cold storage and heart preservation solutions) are enhanced, and novel, promising approaches are emerging.

## 2. Cold Cardioplegia and Ischemic Damage

One of the main limitations in heart transplantation is the problem of viability of the donor heart since there is a certain transport time between donor explantation and recipient implantation, as the use of the donor heart has been limited by the ischemic time of 4 to 6 h [5]. Several techniques have been used to prolong the preservation of the transplantable heart, with perfusion, oxygenation, hyberbaria, and hypothermia being the main ones [6]. Since the first successful heart transplantation in 1967, cold ischemic preservation of the donor heart in a crystalloid solution at 4 °C is the gold standard for myocardial protection, given the fact that this method is inexpensive, easily reproducible, and technically simple [7]. The major advantage of cardioplegia is the reduction in tissue temperature, which leads to decreased oxygen demand and consumption, respectively, resulting in a reduction in myocardial metabolic requirements. Normally, adenosine triphosphate (ATP) is the primary energy source. In prolonged cold storage, ATP is depleted in catabolic action and as a sequala, metabolites are formed, leading eventually to the appearance of necrosis [6].

While cold ischemic storage has been the most widely used technique for preserving the donor heart, there are specific limitations. Prolonged ischemia is one of the main risk factors of early graft dysfunction and therefore primary graft failure. The quality of the donor graft highly depends on the cold ischemic time, which should not be longer than 6 h. Thus, the reduction in this period and the development of better formulations of heart preservation solutions seem mandatory. Prolonged ischemia time has been responsible for early graft dysfunction and primary graft failure, which are the major causes of death during the first month after heart transplantation, with the risk being higher after 180 min of the beginning of cold storage [7]. The duration of myocardial ischemia is an independent predictor of survival up to 15 years after heart transplantation [8]. Poor graft preservation can cause an ischemic injury, which leads to an inflammatory state, responsible for increased percentages of immunorecognition and rejection responses. Recent experimental data showed that the inhibition of inflammation and apoptosis using siRNA could potentially protect donor hearts through the increase in cold preservation time [9]. In addition, there is a report that suggests that the use of the University of Wisconsin (UW) solution may be associated with an increased risk of graft vasculopathy [10]. Several studies demonstrated the potential benefits of decreasing the ischemic time, and some of them suggest that every additional hour of donor heart ischemic time results in a 25% increased risk of death during the first year after the transplant, with a 5% increase thereafter [11].

Current methods of myocardial preservation using cold ischemic storage have multiple negative effects, among which cellular swelling, extracellular edema, and cellular acidosis are the most prominent. In addition, the depletion of metabolic substrate, reperfusion injury, and endothelial damage may be seen. Cellular swelling is triggered by the suppression of the Na-K-ATPase pump and the decrease in the plasma membrane potential, leading to the intracellular entrance of sodium and chloride with water following and causing cell swelling. This phenomenon may be limited by the addition of colloids, anions, and saccharides. Extracellular edema is the result of hydrostatic changes in the organ vasculature. In addition, tissue acidosis is caused by lactic acid production in the context of anaerobic glycolysis attributed to ischemia. Finally, reperfusion injury can be mediated by the production of harmful oxygen free radicals (ROS) and intracellular calcium overload [12].

As mentioned above, cold ischemic storage may be harmful on the coronary endothelium, as both high-potassium-content blood cardioplegic solutions and intracellular crystalloid solutions, which are of ionic composition and are characterized as depolarizing reflecting its high potassium content, can cause chemical damage to the coronary endothelium [8]. Thus, experimental data suggest that crystalloid solutions with low potassium concentrations may lead to the safer preservation of the transplantable heart [13]. Endothelial cell damage might lead to raised capillary permeability and organ edema, the activation of vasoactive compounds and vasospasm, and microvascular hypoperfusion. The overall outcome is early organ dysfunction, graft rejection, and chronic transplant vasculopathy [14]. Furthermore, the viability of endothelial cells is affected by prolonged and severe hypoxia, as apoptotic changes occur. Hypoxia increases the risk of endothelial damage through other mechanisms. Among them, the increase in anaerobic metabolism, the reduction in protein synthesis, and the augmented production of specific interleukins (IL-1 and IL-8), as well as oxygen-derived free radicals have a major role. Endothelial damage can overall cause perivascular and tissue edema, with the increased possibility of microvascular thrombosis and graft dysfunction. On the other hand, hypothermia deliberates the activity of ionic pumps with the loss of transmembrane ionic gradients. This leads to endothelial swelling, as well as to the elevation of the extracellular calcium, with a total effect in the endothelial sensitivity, which is raised [15,16]. Studies showed that the addition of blood or albumin to cardioplegic solutions may improve endothelium-dependent relaxation, causing improved preservation of the donor heart [17].

## 3. Support of the Donor

The optimization of the cardiac donor management through utilization of donor management goals (DMGs) is a key response to the growing need for organs. In fact, this has been shown to improve both the quantity and quality of transplantations through the increased yield of transplants per donor and improved transplant function, respectively [18]. Towards that goal, the evaluation of cardiac function is to be performed right after the donor’s declaration of death by neurological criteria (DNC). Basic components include levels of cardiac markers (CPK-MB and troponin), 12-lead ECG and echocardiogram. Especially important is a thorough transthoracic echocardiogram, as specific measurements have been linked with worse outcomes (e.g., left ventricular wall thickness greater than 1.4 cm and an increased risk of death) [19]. Coronary angiogram is reserved for male donors over 40 and female ones over 45 years old [20].

As per the Canadian Council’s for Donation and Transplantation (CCDT) recommended DMGs, the monitoring and treatment of reversible organ dysfunction should be performed in the ICU environment for 12–24 h [21]. Specific parameters and the corresponding desired goals are described in Table 2. Management challenges are posed by the catecholamine surge and systematic inflammatory response involved in the physiology of death. Cardiac function is affected by the intense sympathetic activation, and contraction band necrosis is possible. Nevertheless, recommended interventions may improve cardiac function and a proceeding echocardiogram with a minimum ejection fraction of 50% is adequate. Electrolyte and fluid imbalance are another common manifestation of DNC, and their management is vital. It is a consequence of posterior pituitary dysfunction and the resultant diabetes insipidus. Thus, management involves vasopressin and desmopressin administration in addition to electrolyte correction [22].

During brain death, both autonomic disturbances and neurohormonal axis failure may be seen, resulting in the rapid decline of circulating hormones. On the other hand, the depletion of myocardial energy stores leads to the predominance of anaerobic metabolism in combination with hemodynamic instability and myocardial dysfunction. Thus, hormonal supplementation, based on the administration of thyroxine (T4), methylprednisolone, vasopressin, and insulin, is widely used as one of the main interventions to improve cardiac function [23]. Hormonal replacement promotes aerobic metabolism and thereby replenish myocardial energy and glycogen reservoir. In this way, the number of the available grafts can be increased, as those grafts previously considered as marginal, now may be added to the transplant quiver. Yet, not all studies support the beneficial effect of hormonal therapy, as there is a concern that an increase in time between brain death and organ procurement can promote inflammation-induced tissue damage. According to current recommendations, hormonal administration has a place in donors with ejection fraction below 45%, whereas overall hormonal therapy is recommended when specific indications are met by the CCDT and “consideration should be given to its use in all donors” [24,25].

Recently, the potential beneficial role of thyroid hormone (TH) in cardioprotection has come to the fore, dispelling the long-held belief that its administration may be detrimental to the ischemic myocardium. First, Pantos et al., showed that TH can be protective in the setting of ischemia–reperfusion injury in a pattern similar to ischemic preconditioning. The possible mechanisms, through which T3 exerts this effect, are the suppression of the pro-apoptotic p38 mitogen-activated protein kinase (MAPK), as well as the increased expression of cardioprotective molecules, such as the heat shock proteins HSP27 and HSP70. Experimental evidence of the administration of T3 after myocardial infarction in rat hearts, showed reduced cell apoptosis, which appears to be mediated by the activation of protein kinase B (Akt) and the miR30a/p53 axis [26].

In addition, T3 may promote the restriction of myocardial injury in the post-ischemic period, contributing decisively to the recovery of graft function, through its binding to the nuclear receptor TRα1, on which T4 has no effect. This feature of T3 is remarkable, since it works as a molecular switch, thereby determining the activity of the receptor. TRα1 receptor expression is increased in the remaining viable myocardium after the ischemic period to levels like those seen in fetal life, in which T3 is very low. The increased concentrations of the hormone and the subsequent binding of the receptor results in the strengthening of the function of the myocardium. This is due to the induction of the maturation and growth of cardiomyocytes, in such a way that T3 can be characterized as a ‘’regeneration hormone’’ [27]. This potential action of the hormone is further strengthened by the experiments of Naqvi et al., who showed that the addition of T3 promotes cell proliferation through the IGF1/Akt pathway [28].

## 4. Protection of the Donor Heart

Reducing occurrence of ischemia reperfusion injury (IRI) of the donor heart is the cornerstone of effective preservation. Current preservation practice is static cold storage (SCS) in heart transplantation solutions (HPSs) following cardioplegia. There is a wide gamut of HPSs currently in use. At least 167 different types of solutions are used, and all of them are classified according to the sodium and potassium anion concentrations, as extracellular and intracellular. Intracellular solutions contain high levels of sodium, potassium, and magnesium, whereas extracellular solutions contain low electrolyte levels [29]. The most frequently used intracellular solutions are the histidine–tryptophan–ketoglutarate (HTK), University of Wisconsin (UW), Euro-Collins, and Stanford (SFT) solutions. On the other hand, Celsior is the commonest extracellular one. Differences in their composition are reflected in their practical and preservation properties. For example, just a single administration of HTK provides 2 h of IRI protection. Celsior contains antioxidants, lactobionate, and glutathione, which protect against oxidative stress during reperfusion. Differences between the two types of cardioplegic solutions are described in Table 3. There are no clear clinical data to support the superiority of one over the others, although there are several smaller studies in animals, suggesting a potential superiority of HTK solution over the other cardioplegic solutions [30]. The continuing development of cardioplegic solutions seems to extend the viability of the donor heart, through certain mechanisms, although there are problems that must be overcome. The commonly used cardioplegic solutions contain various metabolic and oncotic agents, which prevent necrosis and edema formation. However, there are problems related to cellular metabolism in the context of cold storage, organ ischemia, and reperfusion [6].

Despite SCS in HPSs being the main preserving technique in clinical use, research is active in new methods as well as augmenting current ones. Donor heart perfusion via perfusion systems, a method that was used in the early days of heart transplantation, is re-emerging and provides two ways of maintaining oxygenation. One method is performed by providing oxygenated, cold cardioplegic solution (CPS) to the arrested heart. Another, more arduous method is preservation of the beating heart with normothermic oxygenated blood-based buffers [31]. Adding to above technical requirements of this method is the need for viability monitoring and organ function support. Nevertheless, its suitability for organs obtained from circulatory determined death (DCD) donors offset those obstacles. Many anti-ischemic compounds are added to standard HPSs and tested for their contributing effects. Increased effectiveness has been demonstrated using new HPSs, such as the Krebs–Henseleit buffer-based (KHB) and the Dsol solution, which are currently tested [32].

**Table 3 ijms-24-16693-t003:** [33]: Differences between intracellular and extracellular cardioplegic solutions.

Extracellular Solutions	Intracellular Solutions
Composition proportional to extracellular space	Composition proportional to intracellular space
Potassium concentration = 5–30 mmol/L	Potassium concentration = 30–125 mmol/L
Sodium concentration ≥ 70 mmol/L	Sodium concentration < 70 mmol/L
Asystole due to the increased concentration of potassium in the myocardium	Asystole due to the decrease in the concentration difference of ions on either side of the cell membrane of the myocardial cells
Krebs, St Thomas, Celsior	Custodiol, University of Wisconsin, Stanford

## 5. Ex Vivo Machine Perfusion as a Novel Strategy to Preserve the Donor Heart

The sufficient preservation of donor hearts remains a critical issue in cardiac transplantation. The high metabolism of the heart, compared to the other organs, is considered responsible for the rapid depletion of ATP, leading to acidosis, and necrosis of the myocardium, respectively. Ex vivo perfusion appears to be a promising, novel, organ preservation approach to address some of the limitations of cold cardioplegia. This method uses similar solutions to cold cardioplegia, with some modifications, including blood products. Machine perfusion (MP) has the clinical advantage of reducing cold ischemic time significantly in long-distance procurements or complex recipient operations, such as explantation of mechanical circulatory support. Furthermore, MP allows for donor heart assessment before implantation, and this may facilitate the use of marginal donors. More importantly, MP may become a platform for cardioprotective approaches, enabling the repair and reconditioning of donor hearts [34]. In heart transplantation, two methods of ex vivo perfusion have been reported: hypothermic (4–8 °C) and near normothermic (>32.8 °C). Hypothermic ex vivo heart perfusion involves pumping a cold crystalloid solution into the coronary arteries of the arrested heart to deliver oxygen and nutrients while removing toxic metabolites. Several hypothermic ex vivo heart perfusion platforms have been studied or are currently under investigation. Normothermic machine perfusion maintains the donor heart in a warm, contractile, near-physiologic state during transport from the donor to the recipient hospital. The donor heart must be arrested with standard solutions prior to implantation on normothermic machine perfusion [31].

Warm perfusion has demonstrated the enhanced myocardial protection of the donor heart and is used as an alternative option in donor graft preservation. The presence of buffer solutions, metabolic substrates, and hemoglobin (necessary for the transport of oxygen to the tissues) in the blood, makes this method of preservation particularly attractive. As a result, the protection of ischemic myocardium can be provided in a relatively simple way utilizing a nature product, such as blood.

Maintaining donor’s heart through a perfusion platform advocates a plethora of advantages. First, it reduces cold ischemia time, which, if it exceeds 4–6 h, unpreventable increases to the risk of primary graft dysfunction occur, leading to 8% mortality in the first 30 days and post-transplantation mortality at 5 and 15 years, respectively [35,36,37]. Adverse events, such as impaired biventricular function, PGD, prolonged intensive care stay, and the need for prolonged inotropic or mechanical circulatory support, have been associated with protracted cold ischemia time [2,38]. Expanding the donor pool is a primary goal of Ex vivo Heart Perfusion (EVHP). The extension of the transportation time needed from donors to recipient hospital can make a heart transplantation that would have otherwise been excluded due to long cold ischemia time feasible. In 2015, in Australia, researchers have reported a case of 10 h EVHF with the Organ Care System. Even though the recipient required ECMO and inotropic support for 17 h, they were discharged on 15th post-op day with normal cardiac function [39]. The expansion of the donor pool can be achieved by reclaiming hearts for donation after circulatory death [40]. Adequate preservation and recovery of these hearts has been reported with EVHF. Dhital KK et al. were the first who reported three successful cases of heart transplantation with DCD allografts transferred with Organ Care System for preservation, resuscitation, and transportation to the recipient hospital. Within a week, all three recipients had normal cardiac function. In 2017, Dhital also reported 12 cases of EVHF DCD allografts (Maastricht III) with normal biventricular function within the first 30 days [41].

Keeping cardiac allograft perfused may potentially provide valuable time for surgeons to optimize their surgical plan and improve coordination. Patients required explant of left ventricular assist device or redo sternotomy, might benefit from EVHP limiting cold ischemic time, as long as surgeons optimize the recipient for implantation and copying any unanticipated circumstances [40,42]. The ex vivo perfusion of heart allograft provides the opportunity of evaluation and optimization. The metabolic assessment of the allograft depends on lactate levels. Lactate levels higher than 5 mmol/L in samples of aortic root and pulmonary artery cannulas has proved the most powerful predictor of post-transplantation graft failure. Nevertheless, the sensitivity of lactate levels as a post-transplant metabolic marker regarding outcomes remains debatable [43,44].

### 5.1. Normothermic Machine Perfusion: The Organ Care System

The Organ Care System (TransMedics, Inc., Andover, MA, USA) is the only ex vivo heart perfusion platform with warm, oxygenated, and nutrient-enriched donor blood-based buffer currently in use [45]. With the Organ Care System, warm oxygenated red blood cells (RBCs) are pumped into the aorta, perfusing the coronary arteries. The coronary sinus flow then passes through the tricuspid valve and is ejected by the right ventricle into a pulmonary artery cannula, and finally returns to the reservoir. The perfusate typically includes insulin, antibiotics, methylprednisolone, sodium bicarbonate, multivitamins, and fresh donor RBCs. Hemodynamics (coronary flow, mean aortic pressure, heart rate, and electrocardiography), as well as chemical parameters (arterial blood gas, electrolyte, and glucose) are checked regularly during the donor heart transport to the recipient hospital. Target mean aortic pressure is between 65 mm Hg and 85 mm Hg, coronary flow is between 650 mL/min and 850 mL/min, and total arterial lactate less than 5 mmol/L [46]. Those parameters are corrected by adjusting pump flow, maintenance solution rate, epinephrine infusion, pacing rate, and gas flow.

Machine perfusion may become a platform for applying cardioprotective approaches, such as ischemic preconditioning or post-conditioning, and pharmacological cardioprotective agents. In fact, erythropoietin, glyceryl trinitrate, zoniporide, and antioxidants have been used in porcines for ischemic postconditioning and protection against primary graft dysfunction in DCD allografts with positive results [47]. Church et al. used blood and plasma cross circulation with a live paracorporeal animal under anesthesia to consistently support donor porcine hearts during perfusion for 12 h. Similar results have been reported in both groups, and the author concluded that potentially fresh frozen plasma could permit prolonged heart perfusion [48]. Further experiments are needed in chase for the ideal perfusate.

Utilizing the Organ Care System for cardiac allograft preservation demands several resources, leading inevitably to higher cost than standard cold storage techniques. Trained personnel, appropriate equipment, technical support, and harvesting donor blood for priming perfusion platform point to that direction. Nevertheless, this is counterbalanced with expanding of donor pool and the potential cost of primary graft dysfunction. Decreasing cold ischemia time translates to better survival rates. It has been reported that patients had a 25% increased risk of death during first year after heart transplantation for every extra hour of ischemia after 1 h [11]. In addition, it was found that decreased ischemic time by 1 h increases survival by 2.2 years [49]. Familiarization with this technique, as in any new technological improvement, is necessary to consolidate the Organ Care System in routine heart transplantation. Freed et al. have reported their worries in Lancet that the Organ Care System does not seem to provide sufficient clinical benefit to justify the costs [50]. Hitherto, cost-effectiveness analysis is lacking but will be essential for widespread clinical adoption.

Prolonged ex vivo heart perfusion has the disadvantage of the development of myocardial edema, although it is not certain that this worsens the patient’s outcome [51]. Attempts with hypothermic perfusion techniques for preventing edema exist in the literature [52]. Various perfusate techniques have been reported, but further research is ongoing for the ideal perfusion [47]. Optimal evaluation for the cardiac allograft in the perfusion platform at present is limited to biochemical evaluation (mainly lactate) and visual assessment by the surgeon. Extended criteria to widen organ pool by optimization in the Organ Care System are ongoing (Trial to Evaluate the Safety and Effectiveness of The Portable Organ Care System (OCS™) Heart For Preserving and Assessing Expanded Criteria Donor Hearts for Transplantation Heart Pivotal Trial (NCT02323321)).

#### 5.1.1. Experimental Studies with Organ Care System

Hassanein et al. (1998) were the first who proved that the preservation period of a donor heart could be prolonged by normothermic blood-perfusion in the beating working state, preventing ischemia reperfusion injury compared to standard hypothermic arrest and simple storage in a cardioplegic solution [53]. They developed a portable perfusion platform (the precursor of the Organ Care System based on the Langendorff method), which kept swine hearts perfused with oxygenated normothermic homologous blood in a beating state. Hearts were randomized into three groups: for group I (*n* = 5), cardioplegic arrest, 12 h of storage at 4 degrees C with modified Belzer solution, and 2 h of sanguineous reperfusion in the working state; for group II (*n* = 6), 12 h of continuous perfusion in the beating working state, 30 min of arrest (to simulate re-implantation time), and 2 h of reperfusion, as above; and for group III (*n* = 7), coronary ring control hearts. The results revealed that the perfusion group had better left ventricular developed pressure (LVDP) at 2 h of reperfusion, less myocardial edema, and significantly less myocardial acidosis during preservation and reperfusion. This initial evidence supported the hypothesis that a perfusion device could prolong the preservation of the donor heart, by reducing ischemia reperfusion damage and expanding the donor pool.

#### 5.1.2. Clinical Studies with the Organ Care System

PROTECT I was the first clinical trial that evaluated the safety of the Organ Care System in Germany and the United Kingdom. This European study was a prospective single-arm non-randomized safety and performance study conducted between January 2006 and February 2007. A total of 25 donor hearts were included in this trial, with three of them having been excluded due to the assessment criteria. All 20 patients met the primary endpoint of 30-day survival. Following PROTECT I, PROTECT II was conducted in Europe as well but was soon merged into commercial use and no full data collection was published [54]. PROCEED I (2007–2008) is a prospective multicenter trial investigating the safety and effectiveness of the Organ Care System device for cardiac use conducted in US. The primary outcome was 7-day survival, and the secondary endpoints were 30-day patient and graft survival. In total, fourteen hearts met the inclusion criteria, and one was excluded after assessment. A total of 11 out of the 13 donor hearts reached 7- and 30-day survival. However, it was obvious that the strong limitation of small sample size was not adequate for clear conclusions [54].

In 2015, Ardehali A. et al. published in Lancet the results of PROCEED II, a prospective multicenter randomized non-inferiority trial between 2011 and 2013, in Lancet. In total, 130 patients who were to receive donor hearts were randomized into the Organ Care System group or standard cold storage group, correlating with the preservation technique to be used. Primary outcomes were 30-day patient and graft survival with a 10% non-inferiority margin [54]. The 30-day patient and graft survival rates were 94% (*n* = 63) in the Organ Care System group and 97% (*n* = 61) in the standard cold storage, with *p* = 0·45. The secondary results were cardiac-related serious adverse events, severe rejection rates, and the median length of stay in the intensive care unit. Likewise, the results revealed the non-inferiority of the Organ Care System group. The results showed that the donor hearts for heart transplantation were adequately preserved and had similarly short-term clinical outcomes in both groups. The authors acknowledge the potential advantage of the Organ Care System for the expansion of the donor pool with the prolonged adequate preservation of the donor heart for distant areas, but their main aim was the initial observation that Organ Care System is as safe and effective as cold storage for preservation. A remarkable observation of this trial was that the despite total preservation time (out of body) was longer for the Organ Care System group (324 min VS 195 min attributed to the extra time needed to instrument the donor heart into the Organ Care System circuit and optimize the perfusion characteristics), and the mean total cold ischemia time was significantly shorter in the Organ Care System group than in the standard cold storage group (113 min vs. 195 min). Further studies for the metabolic assessment of the allograft are suggested in Organ Care system group. PROCEED II is the first randomized clinical trial to assess the safety and efficacy of an ex vivo heart transplant platform (OCS) in human heart transplantation [55].

Koerner MM et al. (2014) published the results of a prospective, non-randomized, single-institutional clinical study of 159 cardiac allografts transplantations. A total of 29 were assigned for normothermic ex vivo allograft blood perfusion with the Organ Care System and 130 for cold static storage. The primary outcome of the recipient’s survival at 30 days and 1 and 2 years after transplantation suggested similar results (30 days and 1 and 2 years was 96%, 89%, and 89%, respectively), whereas in the CSS group, the survival after oHTx was 95% (*p* = 0.65), 81% (*p* = 0.28), and 79% (*p* = 0.21). The secondary end points were primary and chronic allograft failure, non-cardiac complications, and the length of hospital stay, and only revealed not statistically significant difference. Nevertheless, the authors concluded that normothermic ex vivo allograft blood perfusion in adult clinical orthotopic heart transplantation contributes to more beneficial outcomes regarding recipient survival, the incidence of primary graft dysfunction, and the incidence of acute rejection [56].

The main potential advantage of ex vivo heart perfusion is the expansion of the donor pool, using hearts that are considered unsuitable for transplantation. Normothermic ex vivo machine perfusion permits the functional evaluation of marginal donor hearts. This concept was tested in the EXPAND trial (International Trial to Evaluate the Safety and Effectiveness of The Portable Organ Care System Heart For Preserving and Assessing Expanded Criteria Donor Hearts for Transplantation), which analyzed high-risk transplants with anticipated prolonged ischemic times (>4 h) or marginal donor heart features (left ventricular hypertrophy, an ejection fraction of 40–50%, donor downtime >20 min, and donor age >55 years). The short-term results showed excellent short-term outcomes in 75 patients transplanted from a total of 93 hearts recovered and perfused on the OCS [34].

Sponga et al., in a single-center retrospective study of patients who underwent heart transplantation, also used marginal donors preserved via cold static or warm perfusion. The criteria for marginal donor hearts were age ≥ 55 years; expected ischemic time > 4 h; left ventricular ejection fraction ≤ 50%; interventricular septum thickness ≥ 14 mm; drug abuse history; episodes of cardiac arrest; and the presence of mild coronary artery disease. Ex vivo perfusion allowed for the continuous evaluation of marginal donor hearts, favoring the exclusion of unsuitable grafts and resulted in a reduction in complications and optimal survival up by to 5 years compared to cold cardioplegia. Notably, in this study, histopathological and ultrastructural examination revealed that, even with normothermic perfusion, pathological changes in cardiomyocytes, and endothelium may be seen. Cardiomyocyte degeneration at the end of organ transportation was more prominent in organs preserved with warm perfusion. However, following reperfusion, a significantly higher apoptotic rate was observed in cold-preserved hearts. Morphological changes of endothelial vascular damage were observed when cells were exposed either to cardioplegic or organ preservation solutions. Warm perfusion resulted in a significantly larger proportion of intramyocardial areas of hemorrhage at the end of the perfusion period [57].

Recently, Langmuur et al. published the results of a meta-analysis, including 12 studies of HTx, wherein 260 out of 741 donor hearts were preserved with OCS. Outcomes in the OCS group were equivalent to static cold storage (SCS), regarding either early or late survival outcomes (primary endpoints). Notably, OCS outcomes concerned heart donation both after brain death (DBD) and circulatory death (DCD). The 30-day survival rate was 97.4% for the OCS-DBD group, 96.6% for the OCS-DCD group, and 95.7% for the SCS-DBD group. No differences in short-term survival between OCS-DBD and OCS-DCD, compared to SCS-DBD techniques, were detected in the pooled ORs for 30 d survival. As for the secondary outcomes, no significant differences in the length of hospital stay, primary graft dysfunction, or rejection were present. Interestingly, the pooled 1-year survival was 84.2% for the OCS-DBD group, 89.3% for the OCS-DCD group, and 87.0% for the SCS-DBD group, whereas rates for the pooled 4-year survival was 82.2%, 85.3%, and 80.3%, respectively. However, the main limitation of this meta-analysis was that it mainly involved observational retrospective studies, the majority of which were lacking a randomized and fully comparable control group, likely leading to bias. The authors concluded that the Organ Care System is a safe and an efficient platform for both DBD and DCD donation, especially in advanced surgical procedures or in cases where long-distance transport is required [58]. Machine normothermic perfusion, based on the Organ Care System, is a novel and promising technique in donor heart preservation. Despite the potential beneficial impact, especially in high-risk patients, there is a discordance among the published evidence regarding survival rates in the OCS group compared to static cold cardioplegia. In fact, some studies showed no significant differences between the two groups. [59,60]. However, there is a paucity of long-term outcomes beyond 30 days in most studies. Thus, prospective, randomized trials with a longer follow-up are essential to assess the durability of any benefits. Five clinical trials are currently ongoing, aiming to evaluate potential safety and efficacy of the utilization of the OCS.

### 5.2. Hypothermic Machine Perfusion: Ex Vivo Non-Ischemic Heart Preservation (NIHP)

Non-ischemic heart preservation (NIHP) is an alternative to the Organ Care System heart-graft transport device, providing the ability to procure distant donors while simultaneously rejuvenating marginal heart grafts. In contrast to the OCS, in which a non-cardioplegic warm preservation solution is used, in NIHP, heart perfusion relies on a continuous supply of oxygenated cold cardioplegic solution. The perfusion is enriched with hormones, red blood cells, and nutritional components, maintaining a hematocrit of 8%. With the use of this device, continuous perfusion with oxygenated blood is achieved, while at the same time, the metabolic demands of storage are limited [61].

Data from preclinical studies in porcine hearts have shown that using NIHP, heart preservation can be safely achieved for 24 h, with the advantage of maintaining endothelial function for at least 8 h. In 2020, Nilsson et al., published the results of the first-in-human study, a non-randomized, open-label, phase 2 trial, which showed the safety of NIHP in heart transplantation without heart-related complications or early mortality. When compared to static cold preservation, NIHP appeared to be beneficial in reducing myocardial damage to a greater extent, as well as allograft rejection at 6 months [62].

The main disadvantages of the NIHP method are the appearance of edema, which is often irreversible, as well as the increased incidence of kidney damage in the postoperative period. The possible beneficial effects of NIHP on transplant rejection are also reinforced by Critchley et al., who showed that the use of non-ischemic heart preservation via hypothermic cardioplegic perfusion reduces donor heart immunogenicity. This is ensured mainly through the loss of resident leukocytes, whereas the limitation of the inflammatory response induced by IFN-γ is thought to play a central role [63]. However, in comparison to normothermic machine perfusion, NIHP does not permit the evaluation of the transplant by assessing lactate levels, coronary perfusion pressure, or cardiac contractility. Thus, this preservation method cannot be used to extend the donor pool with marginal grafts or DCD grafts. On the other hand, NIHP seems to be superior compared to EVHP regarding simplicity and safety, especially in cases where machine malfunction may be present [64]. However, the use of machine perfusion, especially for DCD organs, needs more investigation. Furthermore, comparisons between machine perfusion platforms are limited. Head-to-head comparisons would better define the optimal perfusion approach.

## 6. The Role of Triiodothyronine (T3) in Machine Normothermic Perfusion: Future Challenges

Experimental and clinical evidence suggests that early administration, above normal doses, of triiodothyronine (T3) after myocardial infarction appears to be cardioprotective and prevent cardiac remodeling. It is noteworthy that this action of T3 does not entail the worsening of myocardial damage as a sequela of the differential effect of the hormone on healthy and injured myocardium, respectively. The time of administration, as well as the dosage regimen that will be chosen, seem to be decisive parameters in the action of thyroid hormone (TH) in the ischemic myocardium [65]. This may be attributed to the fact that during ischemic stress, a multitude of changes take place, which make TH a major factor for the restoration of cardiac function. Notably, in ischemic myocardium, the reduced conversion of T4 to T3, the increased inactivation of T3, as well as a disturbance in the response of TH receptors to the action of TH may be seen [66]. In addition, early high-dose T3 administration in patients with acute myocardial infarction undergoing primary angioplasty (PCI) prevented cardiac dilatation and remodeling, showed favorable effects on microvascular obstruction, and facilitated infarct healing (ThyRepair Study) [61]. Dose-finding studies could further define the optimal T3 dosage and timing.

ThyRepair was the first trial of its kind to show the potential of thyroid hormone to repair the infarcted myocardium. This may signal a new era for pharmaceutical organ repair. Notably, TH has also been used in cardiac transplantation as an inotrope for supporting donor heart hemodynamics. A meta-analysis of the data provided by several studies showed that TH may not have a significant effect on donor hemodynamics. However, the observation that a favorable TH effect was evident on unstable donors (due to myocardial ischemia) indicates that TH effects are beyond its inotropic action. In accordance, in the analysis of the data obtained from 66.629 donors, the T3/T4 treatment of cardiac donors was associated with the procurement of significant greater numbers of hearts. Furthermore, the effect of the TH treatment was independent of other factors and associated with improved post-transplantation graft survival [67].

In the era of machine perfusion, the cardioprotective effects of TH could be exploited for cardiac donor preservation and repair. Mourouzis et al. showed that the administration of T3 during normothermic perfusion with non-blood crystalloid buffer was shown to have favorable effects on cardiac function and perfusion pressure, and switched death to pro-survival kinase signaling. Especially, the improvement of coronary perfusion pressure with the addition of T3 in the perfusate indicates a favorable effect on microvascular function during ex vivo normothermic perfusion [68,69]. Experimental evidence suggests that T3 exerts its cardioprotective actions likely by suppressing pro-apoptotic signaling, such as p38 MAPK and JNKs, and increasing the activation of cardioprotective molecules like PI3 K/Akt and AMPK. This is thought to be achieved by a delicate balance among pathways, related to apoptosis and survival, respectively (Figure 1). These pathways are suggested to be essential in ischemic injury, and post-ischemic cardiac remodeling.

Mitogen-activated Protein Kinase (MAPK) cascades are key signaling pathways that are involved in the modulation of cellular responses to a wide range of intracellular stimuli, including apoptosis, vascular permeability, cardiac function, immune cell activation, and cytokine production. Thus, MAPKs are responsible for the regulation of multiple cellular functions, such as gene expression, proliferation, and apoptosis. Thyroid hormone seems to promote the inhibition of p38-MAPK, consequently limiting inflammation and apoptosis. In particular, p38-MAPK activation is believed to be implicated in the inflammatory cascade through the stimulation of immune system, resulting in cytokine production [70,71]. Accordingly, the inhibition of p38 MAPK activation results in lower levels of pro-inflammatory cytokines in a brain-dead donor model [72]. In addition, cellular apoptosis as a result of capase-3 activation, and augmented vascular permeability, can be the result of p38 MAPK activation. Gudenwar et al. determined the vital role of p38-MAPK in tissue repair following stress. Therefore, its deactivation is thought to induce proliferation, as well as the regeneration of myocardial tissue [73]. JNKs are another potential target of the TH effect. JNKs may induce apoptosis through two distinct mechanisms of action following their translocation either to the nucleus or mitochondria. In the nucleus, JNKs stimulate apoptosis by augmenting the expression of pro-apoptotic genes, and this is thought to be mediated by the activation of c-Jun/AP1 or p53/73 protein-dependent mechanisms. On the other hand, in mitochondria, JNKs debilitate the anti-apoptotic activity of Bcl-2 superfamily proteins while at the same time inducing apoptosome formation, resulting in the cleavage and degradation of proteins and DNA [74]. Existing evidence supports that JNK inhibition may result in reduced graft rejection and increased graft survival. Nevertheless, recent studies support that the effects of JNK activation on apoptosis vary depending on the activity of other signaling pathways, such as the NF- kB pathway [75].

In addition, T3 facilitates the activation of pro-survival signaling pathways, which in turn are involved in the blockade of cellular apoptosis, as well as the stimulation of angiogenesis. The PI3K-Akt signaling cascade is a signal transduction pathway that enhances survival, angiogenesis, and regulated calcium cycling in cardiomyocytes. T3 may stimulate PI3K signaling by reinforcing the interaction of both TRα and TRβ with the PI3K regulatory subunit p85. Thus, T3 may promote cell survival and the better handling of calcium via Akt activation. Furthermore, enhanced PI3K/AKT/mTOR activity may result in the increased expression of the transcription factor HIF-1α [76,77]. Hypoxia-inducible factor 1α (HIF-1α) promotes cellular response to hypoxia, as well as resistance to apoptosis and the induction of angiogenesis [78]. Furthermore, the PI3K-Akt signaling pathway may ameliorate the expression of other angiogenic agents, such as angiopoietins and nitric oxide (NO). Experimental results support the crucial role of AMP-activated protein kinase (AMPK) in the improvement of left ventricular function, as well as the increase in myocardial tissue viability, following myocardial infarction. As shown, AMPK may augment the phosphorylation of eNOS, resulting in the enhanced bioavailability of nitric oxide (NO). As a result, vaso-protection, angiogenesis, gene transcription, and post-translational modification may be seen [79]. Moreover, AMPK promotes the stimulation of the transcriptional co-activator PGC-1α. PGC-1α promotes mitochondrial biogenesis and improves mitochondrial function in cardiomyocytes in response to conditions that require increased ATP production [80].

Microcirculation is impaired after reperfusion in both acute myocardial infarction and heart transplantation, and this dysfunction is associated with postischemic cardiac remodeling and impaired healing [81,82]. T3 seems to affect microvasculature, debilitating tissue hypoxia microvascular dysfunction following normothermic perfusion. We found that T3 administration after 4 h of normothermic perfusion resulted in a significantly lower coronary perfusion pressure as compared to untreated hearts, indicating the reduced resistance of coronary vessels [69]. Similar observations have been reported in experimental settings of microvascular dysfunction in ischemia–reperfusion injury. Interestingly, T3 may promote survival in response to stress in endothelial cells via the activation of PI3K/Akt [83]. Furthermore, T3 administration prevented cardiac and liver tissue hypoxia in other settings of microvascular dysfunction, such as in experimental sepsis [84].

Mechanical perfusion is now considered as a novel approach in donor heart preservation, contributing significantly to donor’s pool expansion. Experimental evidence supports that the adjunctive use of cardioprotective agents, such as T3, may provide beneficial outcomes in graft preservation, potentially preventing the cardiac remodeling of the donor’s heart. Thus, the addition of T3 to the perfusate may be a promising strategy to optimize donor heart preservation and repair. However, the benefits of T3 treatment are mainly based on animal data, whereas clinical trials testing its use in organ preservation are needed. Despite significant clinical and therapeutical importance, further research is needed on perfusion solutions and pharmacological agents, like T3, to maximize the potential of machine perfusion modalities.

## 7. Conclusions

Despite the advancements in organ preservation, ischemia–reperfusion injury remains an unresolved problem in cardiac transplantation. Using different types of solutions, we have not achieved an ischemia time beyond a few hours. Static cold storage remains the method of choice for organ preservation. However, the method is associated with increased risk of early graft dysfunction, as well as reperfusion and rewarming injury. In an effort to find alternative ways in terms of heart protection and preservation, including extending the donor pool, new methods have been developed. Machine hypothermic or normothermic perfusion devices have been recently used as promising, novel, preservation methods. EVHP seems to be comparable to cold cardioplegia regarding donor preservation and allows for potential repair treatments to be employed and the assessment of graft function before implantation. It is also a promising platform for using marginal organs and increasing donor pool. However, complex logistics, the use of blood-based solutions, and cost-effectiveness remain the serious constrains of this method. In addition, donor selection criteria need to be better defined. Thus, larger randomized, prospective studies are required to validate utility in extended criteria donors. New pharmacological cardiac repair treatments, as well as cardioprotective interventions have emerged and could allow for the optimization of this modality, making it more practical and cost-effective for the real world of transplantation. T3 is known for its favorable effects in donor heart preservation and repair. Cardioprotective features of triiodothyronine are likely to be achieved by its implication in intracellular signaling pathways involving kinases. Nevertheless, the molecular mechanisms of T3′s actions require further elucidation. Therefore, more mechanistic studies would help optimize its clinical use.

## Figures and Tables

**Figure 1 ijms-24-16693-f001:**
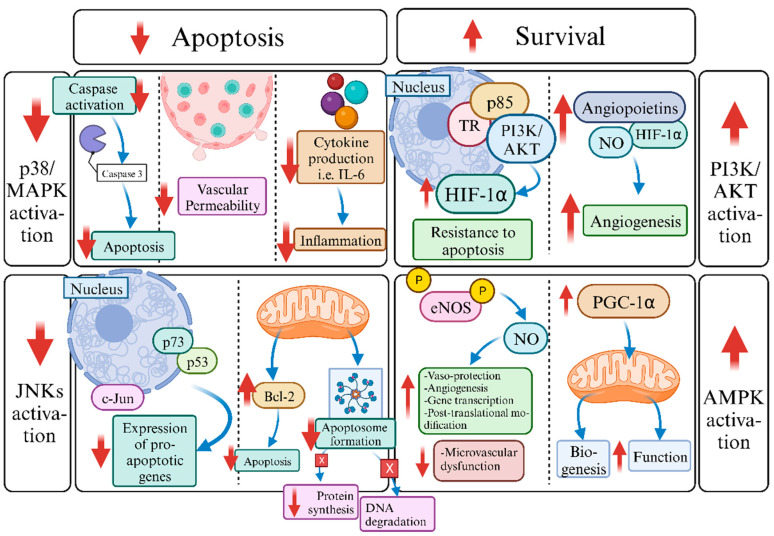
Pathophysiological mechanisms of thyroid hormone (TH) action in normothermic machine perfusion. Abbreviations: p38/MAPK: p38/mitogen-activated protein kinase; IL-6: interleukin-6; c-Jun: c-Jun peptide; JNKs: c-Jun *n*-terminal kinases; Bcl-2: B-cell lymphoma-2 protein family; TR: thyroid receptor; HIF: hypoxia-inducible factor-1α; NO: nitric oxide; PI3K/Akt: phosphatidylinositol 3-kinase/protein kinase B; P: phosphorylation; eNOS: Endothelial nitric oxide synthase; PGC-1α; peroxisome proliferator-activated receptor-gamma coactivator-1alpha; AMPK: adenosine monophosphate-activated protein kinase.

**Table 1 ijms-24-16693-t001:** [4]: Criteria evaluated for marginal graft acceptance according to Massad Cardiology 2004.

Favorable Characteristics	Unfavorable Characteristics
1. Age up to 65 years old 2. Cold ischemia time up to 4–6 h 3. By-passable one- or two-coronary-vessel disease 4. Correctable valvular dysfunction determined by heart ultrasound	1. Prolonged hospitalization 2. History of chest trauma 3. Undersizing or oversizing by more than 20% body weight 4. Open cardiac massage 5. Elevation of myocardial enzyme levels 6. Cardiopulmonary resuscitation >5 min 7. Transient hypotension >30 min 8. High-dose vasopressor requirement 9. Wall motion abnormalities by heart ultrasound 10. Long distance procurement >1000 miles 11. Persistent conduction disturbances

**Table 2 ijms-24-16693-t002:** Standing orders for management of organ donors; ECG: electrocardiogram; PACs: premature atrial contractions; MVO_2_: myocardial volume oxygen; PCWP: pulmonary capillary wedge pressure; SVR: systemic vascular resistance; LVSWI: Left Ventricular Stroke Work Index; NAC: N-acetylcysteine.

Standard monitoring	1. Urine catheter to straight drainage, strict intake and output 2. Nasogastric tube to straight drainage 3. Vital signs every hour 4. Pulse oximetry, 3-lead-ECG 5. Central venous pressure 6. Arterial line pressure 7. Optional PACs
Laboratory investigations	1. Arterial blood gases, electrolytes, and glucose every 4 h and as needed 2. Complete blood count every 8 h 3. Blood urea nitrogen and creatinine every 6 h 4. Urine analysis 5. AST, ALT, bilirubin (total and direct), INR (or PT), and PTT every 6 h
Hemodynamic monitoring and therapy	General targets: heart rate 60–120 bpm, systolic blood pressure > 100 mm Hg, mean arterial pressure ≥ 70 mm Hg. 1. Fluid resuscitation to maintain normovolemia, central venous pressure 6–10 mm Hg 2. If arterial blood pressure is ≥160/90 mm Hg, then: -Wean inotropes and vasopressors -If necessary, start either nitroprusside 0.5–5.0 μg/kg per minute or esmolol 100–500 μg/kg bolus followed by 100–300 μg/kg per minute 3. Serum lactate every 2–4 h 4. Mixed venous oximetry every 2–4 h; titrate therapy to MVO_2_ ≥ 60 mm Hg
Agents for hemodynamic support	1. Dopamine: ≤10 μg/kg/min 2. Vassopresin: ≤2.4 U/h (0.04 U/min) 3. Norepinephrine/epinephrine/phenylephrine (caution with doses > 0.2 μg/kg/min)
Indications for PACs	1. 2-dimensional echo ejection fraction ≤ 40% and/or 2. Dopamine > 10 μg/kg/min and/or 3. Vasopressor support (not including vasopressin if part of hormonal therapy) and/or 4. Escalation of supports
Glycemia and nutrition	1. Routine intravenous dextrose infusions 2. Initiate/continue enteral feeding as tolerated 3. Continue parenteral nutrition if already initiated 4. Initiate and titrate insulin infusion to maintain serum glucose level at 4–8 mmol/L
Fluid and electrolyte targets	1. Urine output 0.5–3 mL/kg/h 2. Serum Na 130–150 mM 3. Normal ranges for potassium, calcium, magnesium, and phosphate
Hematology	1. Optimum hemoglobin: 90–100 g/L for unstable donors, lowest acceptable level is 70 g/L 2. For platelets, INR and PTT, there are no predefined targets; transfuse in case of clinically relevant bleeding 3. No other specific transfusion requirements
Microbiology	1. Daily blood cultures 2. Daily urine cultures 3. Daily endotracheal tube cultures 4. Administer antibiotics for presumed or proven infection
Diabetes insipidus	A. Defined as: 1. Urine output > 4 mL/kg/h associated with 2. Rising serum sodium ≥ 145 mmol/L and/or 3. Rising serum osmolarity ≥ 300 mosM and/or 4. Decreasing urine osmolarity ≤ 200 mosM B. Therapy (to be titrated to urine output ≤ 3 mL/kg/h): 1. Intravenous vasopressin infusion at ≤2.4 U/h and/or 2. Intermittent DDAVP 1–4 μg IV, then 1–2 μg IV every 6 h (there is no true upper limit for dose; should be titrated to desired urine output rate)
Combined hormonal therapy	A. Defined as: 1. T4: 20 μg IV bolus followed by 10 μg/h IV infusion (or 100 μg IV bolus followed by 50 μg IV every 12 h) or T3: 4 mcg IV bolus followed by 3 mcg/h IV infusion 2. Vasopressin: 1 U IV bolus followed by 2.4 U/h IV infusion 3. Methylprednisolone: 15 mg/kg (≤1 g) IV every 24 h B. Indications: 1. 2-dimensional echo ejection fraction ≤ 40% or 2. Hemodynamic instability (includes shock, unresponsive to restoration of normovolemia and requiring vasoactive support (dopamine > 10 μg/min or any vasopressor agent) 3. Consideration should be given to its use in all donors
Heart-specific follow-up	1. 12-lead ECG 2. Troponin I every 12 h 3. 2-dimensional echocardiography: 1. Should only be performed after fluid and hemodynamic resuscitation 2. If ejection fraction ≤ 40% then, insert PACs and titrate therapy to the following targets: 1. PCWP: 6–10 mm Hg 2. Cardiac index: >2–4 L/minute/m^2^ 3. SVR: 80–1200 dynes/sec/cm^−5^ 4. LVSWI > 15 g/kg/min 3. PAC data are relevant for hemodynamic therapy and evaluation for suitability for heart transplantation independent of echo findings 4. Consider repeat echocardiography at 6–12 h intervals 4. Coronary angiography: A. Indications: 1. History of cocaine use 2. Male > 55 years or female > 60 years 3. Male > 40 years or female > 45 years in the presence of 2 or more risk factors 4. ≥3 risk factors of any age B. Risk factors: 1. Smoking 2. Hypertension 3. Diabetes 4. Hyperlipidemia 5. Body mass index > 32 6. Family history of the disease 7. History of coronary artery disease 8. Ischemia on ECG 9. Anterolateral regional wall motion abnormalities on ECG 10. 2-dimensional echo assessment of ejection fraction ≤ 40% C. Precautions: 1. Ensure normovolemia 2. Administer prophylactic NAC, 600–1000 mg enterally twice daily (first dose as soon as angiography indicated) or IV 150 mg/kg in 500 mL normal saline over 30 min immediately before contrast agent followed by 50 mg/kg in 500 mL normal saline over 4 h 3.Use low-risk radiocontrast agent (non-ionic or iso-osmolar), using minimum radiocontrast volume, no ventriculogram

## Data Availability

Not applicable.

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
