# Peer review of "Donor Heart Preservation: Current Knowledge and the New Era of Machine Perfusion"

_ijms, 2023, doi:10.3390/ijms242316693_

Round 1
Reviewer 1 Report
Comments and Suggestions for Authors
The review is interesting but some major point should be adderessed by the authors:
- There is an incongruence between title, abstract and text. In the abstract the role of thyroid hormone are described in detail, but neither the title neither the main body of the text are exclusively focused on this topic.
- The second part of the review is entitled "Cold cardioplegia in graft preservation". It is focused on the risk related to cardioplegia and the possible solutions described in the following parts. A different title better reflecting this topic could result clearer for the readers. However, in general the sequence of the topics reported is not easy to be followed (e.g. the 5. Novel strategies to protect the donor heart vs. cold ischemic damage should imply the sub-parts 5.1 Ex-vivo machine perfusion in donor heart preservation; 5.2 Organ care system. The part limitations of organ care system should be part on the correspondent part).
- In the part 3. the authors describe the role of thyroid hormones supplementation Its description is inadequate, considering the relevance reported in the abstract. Pathophysiological aspects should be described in detail as well as available clinical data.
- In general, for each topi discussed, the molecular mechanisms and the related references should be described more in detail.
- One or more figure summarizing the different aspects discussed in the manuscript could greatly improve the interest of the readers.
Author Response
''Please see the attachment''

Reviewer 2 Report
Comments and Suggestions for Authors
The article reviews current knowledge and novel methods for donor heart preservation and repair in heart transplantation. Prolonged cold ischemic storage is associated with increased risk of early graft dysfunction and primary graft failure. New preservation methods like machine perfusion aim to expand the donor pool and allow continuous graft evaluation and potential repair treatments during transport. Hypothermic and near normothermic machine perfusion may be promising novel methods, allowing longer preservation times and use of marginal donor hearts compared to static cold storage.
The role of triiodothyronine (T3) in donor heart preservation and repair is also discussed. T3 has cardioprotective effects mediated through anti-inflammatory, anti-apoptotic and pro-angiogenic pathways. Experimental evidence suggests T3 administration during normothermic perfusion improves cardiac function and perfusion pressure, likely by suppressing pro-apoptotic signaling and increasing expression of cardioprotective molecules. The addition of T3 to perfusion solutions may be a promising strategy to optimize donor heart preservation and repair. Further research is needed on perfusion solutions and pharmacological agents like T3 to maximize the potential of machine perfusion techniques.
I would the authors to incorporate these following points into the discussion or add more information if available:
- Many of the cited studies on machine perfusion are small animal studies or case reports. Larger randomized controlled trials in humans are needed.
- The benefits of T3 treatment are based on limited animal data. Clinical trials testing T3 in organ preservation are lacking.
- The optimal T3 dosage and timing is unknown. Dose-finding studies are needed.
- The molecular mechanisms of T3's actions require further elucidation. More mechanistic studies would help optimize its clinical use.
- Long-term outcomes beyond 30 days are lacking. Studies with longer follow up are important to assess durability of any benefits.
- Comparisons between perfusion systems (e.g. Organ Care System vs NIHP) are limited. Head-to-head comparisons would better define the optimal perfusion approach.
- Cost-effectiveness analysis is lacking but will be important for widespread clinical adoption.
- The utility of perfusion systems specifically for DCD organs needs more investigation.
- Donor selection criteria for machine perfusion need to be better defined. Larger studies are required to validate utility in extended criteria donors.
Author Response
''Please see the attachment''

Round 2
Reviewer 1 Report
Comments and Suggestions for Authors
I've no further comments.